# Gate-controlled reversible rectifying behaviour in tunnel contacted atomically-thin MoS₂ transistor

Xiao-Xi Li [1,2], Zhi-Qiang Fan[3], Pei-Zhi Liu[4], Mao-Lin Chen[1,2], Xin Liu[5,6], Chuan-Kun Jia[7], Dong-Ming Sun[1,2], Xiang-Wei Jiang[3], Zheng Han [1,2], Vincent Bouchiat [8], Jun-Jie Guo[4], Jian-Hao Chen [5,6] & Zhi-Dong Zhang[1,2]

Atomically thin two-dimensional semiconducting materials integrated into van der Waals heterostructures have enabled architectures that hold great promise for next generation nanoelectronics. However, challenges still remain to enable their applications as compliant materials for integration in logic devices. Here, we devise a reverted stacking technique to intercalate a wrinkle-free boron nitride tunnel layer between MoS₂ channel and source drain electrodes. Vertical tunnelling of electrons therefore makes it possible to suppress the Schottky barriers and Fermi level pinning, leading to homogeneous gate-control of the channel chemical potential across the bandgap edges. The observed features of ambipolar *pn* to *np* diode, which can be reversibly gate tuned, paves the way for future logic applications and high performance switches based on atomically thin semiconducting channel.

---

[1] Shenyang National Laboratory for Materials Science, Institute of Metal Research, Chinese Academy of Sciences, Shenyang 110016, China. [2] School of Material Science and Engineering, University of Science and Technology of China, Anhui 230026, China. [3] State Key Laboratory of Superlattices and Microstructures, Institute of Semiconductors, Chinese Academy of Sciences, Beijing 100083, China. [4] Key Laboratory of Interface Science and Engineering in Advanced Materials, Ministry of Education, Taiyuan University of Technology, Taiyuan 030024, China. [5] International Center for Quantum Materials, School of Physics, Peking University, Beijing 100871, China. [6] Collaborative Innovation Center of Quantum Matter, Beijing 100871, China. [7] College of Materials Science and Engineering, Changsha University of Science & Technology, Changsha 410114, China. [8] University of Grenoble Alpes, CNRS, Institut Néel, F-38000 Grenoble, France. Correspondence and requests for materials should be addressed to D.-M.S. (email: dmsun@imr.ac.cn) or to X.-W.J. (email: xwjiang@semi.ac.cn) or to Z.H. (email: vitto.han@gmail.com)

A decade after the first isolation and study of two-dimensional (2D) materials, their atomically precise integration into van der Waals (vdW) planar heterostructures[1, 2] is now forming an outstanding platform for developing novel nanoelectronic devices[3–5]. Such platform has been the source of many recent advances in electrical engineering that takes the advantages of the coupling of mono- or few-layered two-dimensional (2D) materials such as graphene, hexagonal boron nitride (h-BN), and transition metal dichalcogenides (TMDCs). It has thus far thrived a rich variety of physical phenomena, including metal oxide semiconductor field effect transistors (FETs)[1], spintronics memory devices[6], photovoltaics[5], and atomically thin superconductors[7]. Although doping control by an electrostatic gate in those devices has enabled tremendous opportunities, the lack of gapped 2D channel with complementary (p and n) polarities has hampered its application in logic units based on the co-manipulation of diodes and field effect transistors, each has been the core of modern electronics. $MoS_2$ is among the most studied TMDC compounds for both its outstanding electronics and optoelectronics properties as it combines well-defined bandgap, stability in ambient conditions and relatively high charge carrier mobility. Indeed 2H-type molybdenum disulfide ($2H-MoS_2$) has a thickness-dependent bandgap of 1.3 eV indirect gap ~1.9 eV direct gap from bulk down to single layer, respectively[8]. It therefore holds great promise not only for fundamental studies[7, 9, 10], but also for future applications such as high performance FETs and opto-electronics[11–17]. Field effect transistors involving atomically thin $MoS_2$[11] as the active channel have enabled original architectures which unlock new features such as sub-thermionic inter-band tunnelling exhibiting unprecedented minimum sub-threshold swing[12], or ultra-short gate-length FETs[13], opening promising pathways for further enhanced integration.

To fulfill the desired performances of CMOS-type logics using $MoS_2$ FETs, one of the key (yet evasive) goals has been achieving programmable ambipolar operation (i.e., obtaining easily reconfigurable same-chip n- and p-doping in $MoS_2$ FETs). However, to date, only few experiments[18–20] reported hole transport in $MoS_2$, which was achieved through gate dielectric engineering with high gate voltage operation[19] or in an ionic liquid gating environment[20]. Great efforts have been conducted to pursue ambipolar field effect and further gate tunable rectifying characteristics in $MoS_2$ based heterostructures, including $MoS_2$ coupled with other materials such as carbon nanotube films[21]. Similar effects can also be found in n-type TMDCs vdW interfaced with p-type TMDCs[22, 23] or with organic crystal thin films[24].

Here, we show an alternative route based on architecture-engineering: on the basis of the well known technique of vdW heterostructure but with a crucial refinement of the so-called reverted transfer, we enable the fabrication of very reliable high quality h-BN tunnel barriers which gives rise to gate tunable rectification and reversible pn to np diode behaviuor in tunnel-contacted few layer $MoS_2$ transistors.

## Results

**h-BN as an ultra-thin dielectrics for carrier injection via tunnelling.** In this work, we demonstrate the design and room temperature operation of FETs based on a tunnel-contacted (TC) $MoS_2$ channel. The tunnel barrier insulating layer is implemented by an ultra-thin capping layer that enables the vertical tunnelling of electrons from the top deposited electrodes. Ultra-thin (one or few monolayer) BN has been identified in the past as an efficient dielectric essential to a number of vertical transport devices, including graphene tunnel transistors[25–30], and excitonic superfluid double layer systems[31, 32].

As the few-layers h-BN is used as the top most layer, it assumes the role of an atomically uniform potential barrier, across which electrons are coupled through the tunnelling process. For that purpose it is required to be contaminant- and wrinkle-free.

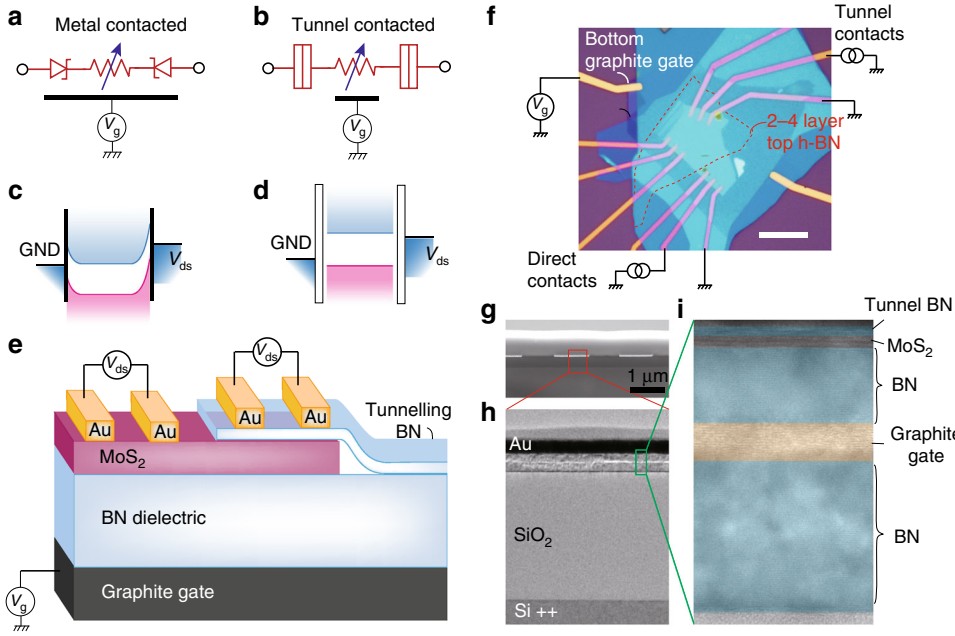

**Fig. 1** Comparison between metal-contacted and metal/insulator tunnel-contacted $MoS_2$ FETs. **a** Schematics of a metal-contacted $MoS_2$ film leading to a Schottky barrier field effect transistor (SB-FET). **b** Schematics of a tunnel-contacted $MoS_2$ field effect transistor (TC-FET). **c**, **d** Semiconductor representation of the energy levels respectively for SB-FET and TC-FET showing the absence of band bending in TC-FETs. **e** Schematics cross section of the device showing SB-FET and TC-FET side by side on the same $MoS_2$ flake. **f** Optical micrograph of a typical TC-FET device. Red dashed line highlights the two to four layer tunnel top h-BN, which covers half the $MoS_2$. Scale bar is 10 μm. **g** Scanning electron microscopy (SEM) image of the cross-section of the graphite-gated $MoS_2$ vertical tunnel device, with its boxed area zoomed in transmission electron microscopy (TEM) images in **h**, **i**

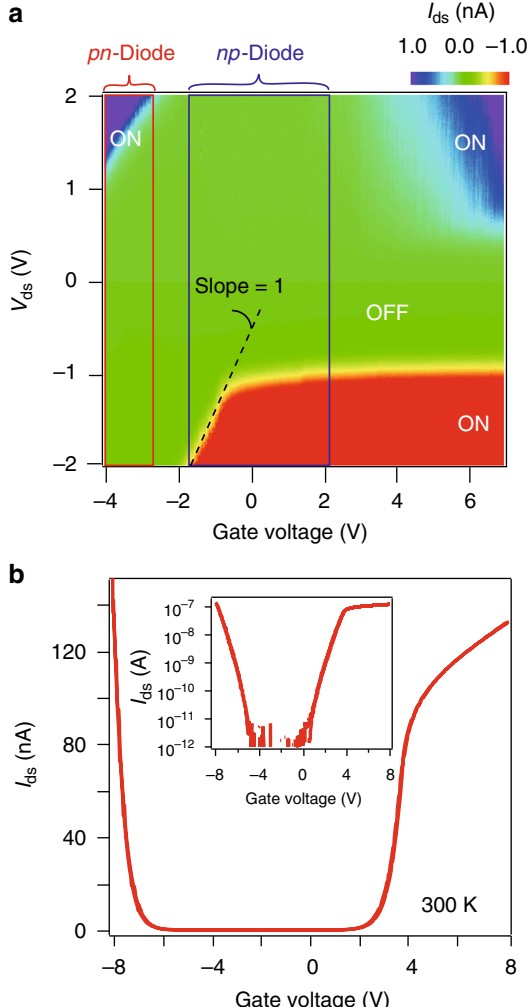

**Fig. 2** Transport characteristics of MoS$_2$ TC-FETs. **a** Color map of output curves ($I_{ds}$ vs $V_{ds}$) at different gate voltages for a typical tunnel-contacted device (room temperature operation). Red and blue-boxed areas highlight the operation range in gate voltage for *pn* and *np* diodes, respectively. **b** Typical ambipolar field effect curve at $V_{ds} = +2$ V measured in samples fabricated by the reverted vdW stacking method. Inset: same data in a semilog plot

Recent results on shot noise measurements in metal–hBN–metal tunnel junctions confirm that h-BN behaves as an ideal tunnel barrier[33].

With the conventional scheme of metal/MoS$_2$ contact, Fermi level pinning at the contact interface usually leads to a gate-dependent Schottky barrier (SB)[34], which results in extra contact resistance that interferes with device performance (Fig. 1a, c). Here, by using the reverted vdW stacking method (Supplementary Figs. 1 and 2 show the experimental details of the reverted vdW process), large-area and wrinkle-free few-layered h-BN can be inserted between metal contacts and 2D semiconductor channel. We found that the presence of tunnel barriers in the form of two to four layer h-BN can suppress the SB, and chemical potential of the MoS$_2$ layer can be adjusted in a uniform manner across the entire channel, achieving precise electrostatic control of the Fermi level of the 2D layer (Fig. 1b, d). Ambipolar field effect at finite source–drain bias, and consequently fully reversible *pn* to *np* diodes by gating was obtained.

Schematic together with an optical image of a typical TC device is shown in Fig. 1e, f. Same flake of few-layered MoS$_2$ is contacted by normal metal contacts, and TC electrodes. Atomic force microscopy image confirms that devices made by our reverted vdW stacking method exhibit atomically flat top tunnel layer, which is free of wrinkles nor ruptures over $10 \times 10 \, \mu m^2$ area (Supplementary Fig. 3). The cross-sectional transmission electron microscopy specimen prepared by focused ion beam of the sample in a local area under metal electrodes is shown in Fig. 1g–i. Typical width of the electrodes are around $1 \, \mu m$, with the MoS$_2$ channel beneath having dimensions $L \times W$ of $1 \, \mu m \times 1$–$5 \, \mu m$ for the tested devices. The multi-layered vdW heterostructure can be clearly seen with a tunnel h-BN on top of few-layered MoS$_2$. To improve the gate efficiency and uniformity[35], graphite flakes with thickness of about 4–6 nm are used as electrostatic gate spaced by a ~10 nm h-BN under the MoS$_2$ channel (Fig. 1i).

**Ambipolar field effect at certain bias condition in MoS$_2$ TC-FETs.** First, we characterize the MoS$_2$ FET with conventional Au (50 nm) electrodes. As shown in Supplementary Fig. 4, transport measurements of them show typical n-type FET behaviour. Color map of *IV* characteristics at fixed gate voltages ($V_g$) indicates ON states at positive and negative bias voltages ($V_{ds}$) on the electron side, while the channel turns off on the hole side. *IV* curves at fixed $V_g$ slightly deviate from linear behaviour, while the transfer curves at fixed bias voltage $V_{ds}$ show typical *n*-type unipolar field effect (Supplementary Note 1). These behaviours are standard in MoS$_2$ FET, agree with previously reported[9, 11, 15].

A striking consequence of the insertion of an ultra-thin h-BN below metal contact is the dramatic change in the color map of *IV* curves at fixed gate voltages, as shown in Fig. 2a. Instead of the rather symmetric $V_{ds}$ polarization with ON state only seen in the electron side for metal-contacted MoS$_2$ FET, the vertical TC-FET on the same piece of MoS$_2$ flake, as well as in the same gate range, features strongly asymmetric $V_{ds}$ polarization in the whole gate range. Surprisingly, when $V_{ds}$ is larger than a threshold value of about 1 V, the device starts to exhibit ambipolar transfer curves, with ON state observed on both electron and hole sides at source–drain bias above +1 V. A typical such ambipolar field effect curve is shown in Fig. 2b. A detailed comparison of transfer curves between MoS$_2$ normal FET and TC-FET, as well as data from various samples are given in Supplementary Figs. 5–7 and Supplementary Note 2. We note that recent report shows that a monolayer chemical vapor deposited h-BN spacing layer can diminish SB at the metal contact, giving rise to a tripled output current in the transistor[36]. However, we did not see such behaviour in our vertical tunnel devices, which may be a result of the less-defected and thicker tunnel h-BN crystals used in this work.

To better understand the obtained result in Fig. 2a, we now plot the line cuts of *IV* along fixed $V_g$. It is found that, at the largest negative gate voltage of about −3 V (all $V_g$ and $V_{ds}$ in the measurements were pushed to the limit which keeps gate leakage negligible), the output curves behave as typical *pn* diode with rectification characteristics (Fig. 3a), and on/off ratio over $10^5$ (Fig. 2b). When gate voltage is brought into the range of −1 to +2 V, it is seen that the diode behaviour is inverted into *np* type by solely tuning the gate (Fig. 3b). The ON side is now in the negative bias voltage direction, as marked by boxes in Fig. 2a. Upon further doping to the electron side, i.e., at larger positive gate voltages, the output curves gradually shift from the diode behaviour into an asymmetric *IV* with the low bias range following the conventional semiconducting trend, but rather

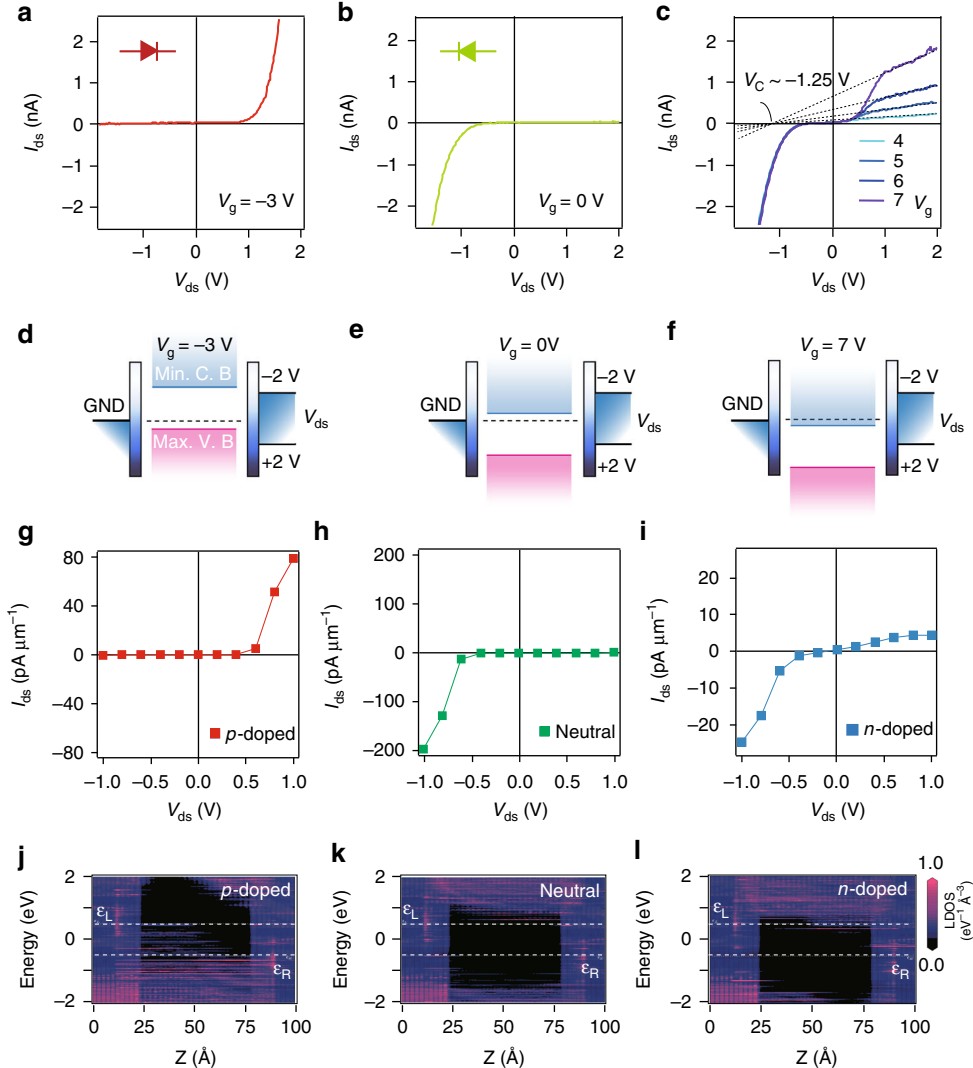

**Fig. 3** Room temperature gate-controlled reversible rectifying diode in a TC-FET. **a–c** *IV* curves showing perfect rectifying behaviour with reversible polarity characteristics of MoS$_2$ TC-FETs. **a–c** are linecuts of Fig. 2a, with output curves along fixed gate voltages of −3, 0, and 4–7 V, respectively. While **d–f** are the corresponding schematic band alignment pictures. **g–l** Simulations of rectifying characteristics of tunnel-contacted MoS$_2$ FET. **g–i** Simulated *IV* characteristics of the MoS$_2$ vertical tunnel FET at hole doping, neutral, and electron doping, respectively. At these corresponding doping level, their simulated PLDOS at $V_{ds} = +1$ V are shown in **j–l**

linear at large positive bias. Strikingly, the linear parts can be extrapolated into a single crossing point on the zero-current axis, with a crossing voltage $V_C$ of about −1.25 V (Fig. 3c). This extrapolated crossing point of *IV* curves is not readily understood and provides food for further experimental and theoretical studies.

## Discussion

We propose a simple band alignment model to explain the observed behaviour of gate-induced switching between *pn* to *np* diodes. In conventional metal-contacted MoS$_2$ devices, due to the work function mismatch, SB forms at the interface of metal and 2D materials, as a result of Fermi level pinning and band bending near the interface (Fig. 1a, c). However, tunnel h-BN in our case overcomes this problem, leading to a relatively free moving conduction and valence bands (Fig. 1b, d). At each stage of electrostatic doping in Fig. 3a–c, Fermi level sits at a fixed energy between the minimum of conduction band and the maximum of valence band, respectively. This free band alignment model offers

a good description of the *pn* to *np* diode inversion in a $V_{ds}$ range of ±2 V, as illustrated in Fig. 3d–f. Moreover, when Fermi level enters conduction band from the band gap, a slope of unity in $V_{ds}$ vs $V_g$ can be extracted in Fig. 2a in the negative $V_{ds}$ regime, indicating a strong energetic coupling of chemical potential from the electrostatic gate to the electronic band in the few-layered MoS$_2$ channel. Once the Fermi level enters the conduction band, the gate becomes capacitively coupled owing to the large density of states, giving rise to a significantly reduced slope of $V_{ds}$ vs $V_g$.

In the following, we compare the measured data with first-principles simulations. For simplicity, we consider the simplest scenario of monolayer MoS$_2$ tunnel device with a channel length of about 6 nm and two-layered tunnel h-BN (computational details can be found in Supplementary Figs. 8–11 and Supplementary Note 3). Compared to Fig. 3a–c, first-principles calculations based on the simplified model give qualitative agreement with experimental observations. As shown in Fig. 3g–i, the two-layered h-BN TC MoS$_2$ FET in our calculated model shows *pn*, *np*, and asymmetrical full pass rectifying characteristics at hole doping, neutral, and electron doping, respectively.

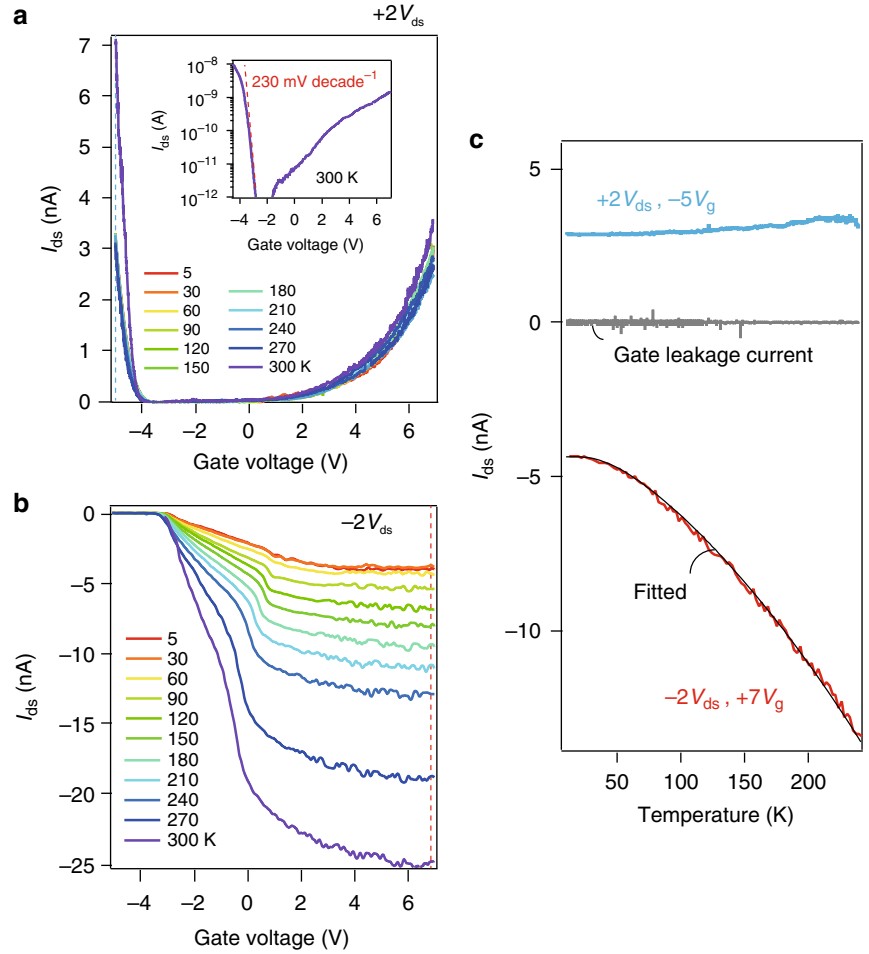

**Fig. 4** Temperature dependence of transfer curves in a $MoS_2$ TC-FET. **a**, **b** Transfer curves at different temperatures for the device shown in Fig. 2, at drain source voltages $V_{ds} = +2$ and $-2$ V, respectively. Inset in **a** is a log scale of the field effect curve. **c** Line traces of temperature dependence of $I_{ds}$ at fixed gate voltage along the blue and red dashed lines in Fig. 5a, b, respectively. Gray solid line indicates the gate leakage current during the same measurement. Solid fitting line in Fig. 5c is fitted using Eq. (1) in the main text

Their corresponding projected local density states (PLDOS) at $V_{ds} = +1$ V are shown in Fig. 3j–l. One can see in the LDOS that the effective transmission forbidden region $\Delta$ in TC device is about 2.5 eV, which is largely enhanced due to the existence of h-BN tunnel barrier ($\Delta \sim 1.8$ eV in normal contacted device, shown in Supplementary Fig. 10). The simulated results echo our hypothesis of free band alignment model in Fig. 3d–f. Fermi level pinning in metal-contacted devices are suppressed by ultra thin tunnel contact, resulting in the observed finite-bias ambipolar field effect, as well as gate tunable rectifying characteristics with multiple operation states.

It is of fundamental interest to study the temperature dependence of tunnelling current in the $MoS_2$ TC-FETs. Figure 4a, b plots the transfer curves of the same device in Fig. 2a, with $V_{ds} = \pm 2$ V at different temperatures from 300 K down to 5 K. It can be seen in Fig. 4a that bipolar transfer curves at $V_{ds} = +2$ V show very weak temperature dependence. A plot of the transfer curve at 300 K is plotted in the inset of Fig. 4a, the sub-threshold swing is extracted on the hole side to be about 230 mV decade$^{-1}$, higher than the 60 mV decade$^{-1}$ theoretical limit[37]. On the contrary, at $V_{ds} = -2$ V, the transfer curves show rather strong temperature dependence (Fig. 4b), with the $I_{ds}$ decreasing upon lowering the temperature. Single traces of $I_{ds}$–$T$ monitored at $+2V_{ds}$ with $-5V_g$, and $-2V_{ds}$ with $+7V_g$ are plotted in Fig. 4c, colors are picked according to the dashed lines in Fig. 4a, b, respectively. The

negatively source–drain biased $I_{ds}$–$T$ curve at $+7V_g$ (red curve) can be fitted by a phonon-assisted tunnelling model[38]:

$$I \propto \frac{eE}{(8m^*\varepsilon_T)^{1/2}}[\Omega - \gamma]^{1/2}[1 + \gamma^2]^{-1/4}$$
$$\exp\left\{-\frac{4}{3}\frac{(2m^*)^{1/2}}{eE\hbar}\varepsilon_T^{3/2}[\Omega - \gamma]^2\left[\Omega + \frac{1}{2}\gamma\right]\right\}, \quad (1)$$

where $\gamma = \alpha\sqrt{2m^*/\varepsilon_T}\frac{\hbar\omega^2}{eE}\left(2[\exp(\hbar\omega/k_BT) - 1]^{-1} + 1\right)$, and $\Omega = (1 + \gamma^2)^{1/2}$, with $\alpha$ being a fitting parameter, $E$ the electrical field strength, $\varepsilon_T$ the tunnel energetic depth, $m^*$ the electron effective mass, $\hbar\omega$ the energy of the phonon taking part in the tunneling process, $e$ and $k_B$ the element charge and Boltzmann's constant, respectively. Using an effective mass of about $0.018m_e$[39], the best fit in the black solid line Fig. 4c gives $\varepsilon_T = 0.6$ eV and $\hbar\omega \sim 11$ meV.

Finally, as a proof of principle for realizing gate-tunable rectifier in the $MoS_2$ TC-FET, we used a simple diode circuit with load resistor of 1 M$\Omega$ and output to a 100 M$\Omega$ impedance voltage amplifier (1× amplification was used in the measurement), as illustrated in the schematics in Fig. 5a. As seen in Fig. 5b, when a sinusoidal wave is input in the $MoS_2$ TC-FET, output wave starts from a positively rectified half wave in the largest hole doping side, and can be first gate tuned into an intermediate OFF state,

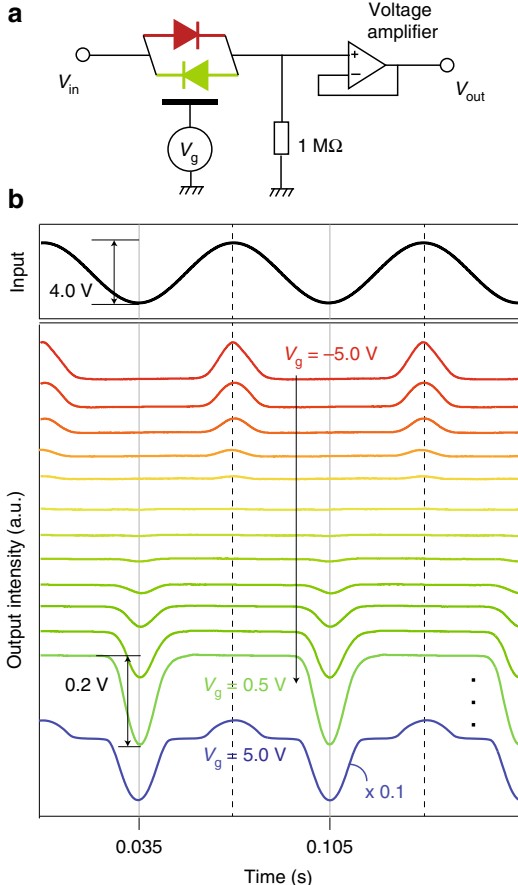

**Fig. 5** Reversal rectification of an analog harmonic signal in MoS$_2$ TC-FET. **a** Schematics of the gate-control rectifier device placed in a measurement and biasing circuit. The MoS$_2$ TC-FET is symbolized as a polarity-switchable diode. **b** Input (harmonic signal ~13 Hz) and output waves of the gate-controlled diode. A $\pi$ phase shift, together with multiple states of output level (e.g., pn diode, OFF, np diode, and full pass), in the rectified output wave can be seen via gating. Each measured curve was averaged over 150 recorded traces

followed by a negatively rectified half wave in the electron doping side. Further electron doping recovers both positive and negative half output wave, with different amplitude. This gate-tunable rectification inversion with a $\pi$ phase shift phenomenon, together with multiple states of output level (e.g., pn diode, OFF, np diode, and full pass), has not been reported before, and can be of great use in future gate-tunable logic circuits with atomically thin conduction channels. As discussed in Supplementary Note 4, it is noteworthy that in a device directly fabricated on SiO$_2$, we obtained a cut-off frequency in such MoS$_2$ TC-FET of about 20 kHz when the Si gate is heavily doped (Supplementary Fig. 12). Moreover, stability and reliability in 2D materials based devices have been a timely topic[40, 41], which is crucial from the application point of view. For example, the thin h-BN layer intercalated between the metal contacts and MoS$_2$ channel can cause extra charge trapping that may lead to inferior reliability as compared to conventional metal-contacted MoS$_2$ FETs (Supplementary Note 5). We rule out this possibility based on the hysteresis measurements, as shown in Supplementary Fig. 13.

To conclude, we have developed a reverted vdW stacking method for high yield fabrication of resist-free pristine vdW heterostructure with ultra-thin top layer. This method itself opens

new routes to a number of applications such as scanning tunnelling microscope on pristine 2D materials supported by another, as well as the high quality spacing layer for tunnelling electrodes. Using this technique, we have demonstrated a vertical TC MoS$_2$ transistor, in which suppression of band bending and Fermi level pinning is realized. The so called TC Field Effect Transistor hence gives rise to gate tunable rectification with fully reversible pn to np diode, leading to multiple operation states of output level (e.g., positive-pass, OFF, negative-pass, and full-pass). The observed ambipolar field effect at finite positive $V_{ds}$ shows on/off ratio up to $10^5$ in such MoS$_2$ FETs, with an output current reaching the order of 100 nA on both electron and hole sides. We proposed a free moving band alignment model to explain the behaviour of the MoS$_2$ TC-FET, which is further qualitatively supported by a simplified first-principles simulation model. This work paves the way for future application in gate-tunable logic devices with atomically thin semiconducting channels.

## Methods

**Reverted vdW heterostructures fabrication process.** In order to have resist-free pristine vdW heterostructures, one of the limitations is its stacking sequence: a thick enough h-BN has to be picked up first by polymer (Propylene-Carbonate, PPC, for example) to serve as a top layer. When the top layer is too thin (<5 layers), ruptures and wrinkles increase significantly, thus reduce the quality of the final device. We solved this problem by developing a reverted vdW stacking method: few-layered MoS$_2$ is sandwiched by a thick (~10 nm) BN (crystals from HQ Graphene) and thin (two to four layer) BN, respectively, with the resulted top later picked up lastly (Supplementary Methods). vdW heterostructures were fabricated using an integrated system E-Stack-One from Eoulu Co., Ltd., Suzhou, China. When the whole stack is collected, the PPC stamp will be flipped upside down, peeled off with care from the PDMS substrate, and slowly landed onto a hot plate of about 100 °C (Supplementary Methods). At this stage, the stack will be floating on the PPC film, which can be completely evaporated in a vacuum annealer at 350 °C for around 20 min. Followed by standard lithography and metallization. MoS$_2$ flake is half covered by two to four layer h-BN, and Au electrode with thickness of 20 nm is deposited onto the stack, forming conventional direct contacts and tunnel contacts, respectively. Electronic transport was measured on a Cascade probe station at room temperature, and in a Quantum Design PPMS system with a home-made sample probe interfaced with external measurement setup at low temperatures, respectively.

**First-principles simulations.** The device simulations in this work are carried out by using the first-principles software package Atomistix ToolKit, which is based on density-functional theory in combination with the non-equilibrium Greens function[42]. The exchange-correlation potential is described by the local density approximation and the wave function is expanded by the Hartwigsen–Goedecker–Hutter (HGH) basis in this work. More computational details are discussed in Supplementary Note 3. The real space grid techniques are used with the energy cutoff of 150 Ry in numerical integrations. The geometries are optimized until all residual force on each atom is smaller than 0.05 eV Å$^{-1}$. The current can be calculated by the Landauer formula[43]:

$$I(V_{ds}) = \frac{2e}{h} \int T(E, V_{ds})[f_S(E, V_{ds}) - f_D(E, V_{ds})]dE. \quad (2)$$

Here, $V_{ds}$ is the bias voltage between the drain and the source, $T(E, V_{ds})$ is the transmission coefficient, $f_S(E, V_{ds})$ and $f_D(E, V_{ds})$ are the Fermi-Dirac distribution functions of the source and drain, respectively. The transmission coefficient $T(E, V_{ds})$ as a function of the energy level $E$ at a certain $V_{ds}$ can be calculated by the formula:

$$T(E, V_{ds}) = \text{Tr}\left[\Gamma_S(E)G^R(E)\Gamma_D(E)G^A(E)\right], \quad (3)$$

where $G^R(E)$ and $G^A(E)$ are the advanced and retarded Greens functions of the scattering region, respectively.

**Data availability.** The data that support the findings of this study are available from the corresponding authors upon reasonable request.

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

## Acknowledgements

This work is supported by the National Key R&D Program of China (2017YFA0206302), the National Basic Research Program of China (973 Grant Nos. 2013CB921900, 2014CB920900), and the National Natural Science Foundation of China (NSFC, Nos. 11504385 and 51627801). X.-W.J. acknowledges supports by the NSFC Grant 11574304, Chinese Academy of Sciences-Peking University Pioneer Cooperation Team (CAS-PKU Pioneer Cooperation Team), and the Youth Innovation Promotion Association CAS (grand 2016109). D.-M.S. thanks the NSFC grant 51272256, 61422406, 61574143, and MSTC grant 2016YFB04001100. X.L. and J.-H.C. acknowledges support from the NSFC Grant 11374021 and 11774010. Z.-D.Z. acknowledges supports from the NSFC with grant 51331006 and the CAS under the project KJZD-EW-M05-3. V.B. acknowledges support from the EU FP7 Graphene Flagship (project no. 604391), J2D project grant (ANR-15-CE24-0017) from Agence Nationale de la Recherche (ANR), and the Hsun Lee Award program of the Institute of Metal Research, CAS. The authors are grateful for helpful discussions with Prof. Benjamin Sacépé, Prof. Vasili Perebeinos, Prof. Antanas Kiveris, and Prof. Ji Feng.

## Author contributions

Z.H. and Z.-D.Z. conceived the experiment and supervised the overall project. X.-X.L. fabricated the samples. X.-X.L., V.B., and Z.H. carried out experimental measurements; D.-M.S. and M.-L.C. provided clean room support for the experiment; X.L. and J.-H.C. contributed to electron beam lithography and device fabrication. X.-W.J. and Z.-Q.F. conducted the theoretical simulations. J.-J.G. and P.-Z.L. carried out the TEM characterizations. Data analysis and interpretation were done by Z.H., X.-X.L., V.B., and C.-K.J.; the manuscript was written by Z.H. with discussion and inputs from all authors.

## Additional information

**Competing interests:** The authors declare no competing financial interests.

