## [Peer Review File · Nature Communications]

Reviewers' comments:

Reviewer #1 (Remarks to the Author):

The authors present experimental techniques to 1) produce wrinkle-free hBN layers between metals and MoS₂ and to 2) show that these layers can serve as tunneling barriers that suppress Schottky barrier formation and Fermi level pinning in MoS₂ based transistors. Their tunneling contact devices show gate tunable doping and ambipolar diode behavior.

To my understanding, these findings are indeed novel and of great importance for TMD based device designs. Being a "pure theoretician", I cannot judge the experimental techniques presented in this work. However, I agree with the authors that their findings of how to fully gate control the effective charge density in MoS₂ based FETs has high impact on the overall TMD-based device literature. In that light, I believe the publication of this work in Nature Communications is appropriate after mandatory changes are done: I have some concerns about the theoretical section of this work:

It would be interesting to see what effective barrier height the DFT calculations showed for the hBN barrier. The observed current densities of the tunnel-contact FETs are 2-3 orders of magnitude smaller than the conventional metal contacts. Do (even) thinner barriers show higher current densities with still ambipolar characteristics? Do the theoretical calculations show saturation of the current at higher voltages?

To make the DFT results reproducible, however, the authors have to add more information about the relative orientation of the hBN, MoS₂ and metal. The relaxation convergence limit alone is not a unique information for the structure setup otherwise. How is the structure setup in the model? Is that relaxation limit small enough for converged results? How much strain is in the system with the achieved relaxation result. What is the system's size in periodic direction? Is the energy cutoff of 150Ry a converged value?

I am also missing a justification for the chosen LDA functional. If that was for convenience/numerical load reduction only (irrespective of possibly wrong band gap results) but to get qualitative comparisons, it should be mentioned in the paper.

Reviewer #2 (Remarks to the Author):

The authors report a very interesting realization of MoS₂ TC FETs which allows to suppress the Schottky barriers and Fermi level pinning by using tunnel thin hBN underneath S/D contacts. Under some bias conditions this brings to the p-FET performance, which has been previously considered as not typical for MoS₂ FETs. The results are properly verified by comparing the performance of MoS₂ TC FETs and conventional (SB) MoS₂ devices fabricated on the same chip. However, the paper can be considerably improved by addressing the following points before publishing in Nature Communications.

Major comments:

1) The use of tunnel-thin hBN layer used for the passivation of the S/D contacts introduces an additional MoS₂/hBN interface into the device architecture. This may lead to additional reliability issues due to charge trapping by hBN defects which can exchange charges with the MoS₂ channel. Thus I would suggest the authors to verify the presence of the hysteresis on the I_d - V_g and, most importantly (to understand the contribution of the tunnel layer), the I_d - V_d characteristics of MoS₂ TC-FETs. Comparison of the hysteresis results for TC and standard (SB) MoS₂ FETs could be provided in the Supporting Information.

2) In the end of page 5 the authors say that the leakage currents (through the thin hBN layer) are limited by 100pA. However, in Fig.S6 the tunnel current can be as high as 12nA. How one should understand this? Also, it seems that both the useful (field-effect) and leakage currents are called I_d s in the graphs and in the text. To avoid any confusion, can the authors call the leakage current as I_{leak} (for instance)?

3) A detailed discussions of the Figures from the Supporting Information (e.g. pages 5-6) makes the paper less comprehensive. I would suggest to move these details into the SI (where these

figures are provided), while leaving only brief references to the SI in the main text.

4) One of the main advantages of MoS₂ compared to other TMDs is large I_{on} values. However, in the TC MoS₂ FETs I_{on} is nearly 3 orders of magnitude lower than in conventional MoS₂ devices (nA vs. uA). As I understand, this is because of "screening" of the applied V_d due to the presence of the insulator under the S/D contacts (e.g. 1V V_ds in TC FETs could be equivalent, for instance, to 0.1V in conventional devices). At the same time, the major problem of these devices seems to be a considerable limitation on V_d operation range (i.e. one can not apply very high V_d in order to avoid damage of thin hBN layer). Can the authors comment on this and estimate "critical" V_d for the hBN thickness used?

Minor Comments:

1) In Fig.S9 caption the authors say that the bandgap of monolayer MoS₂ is 1.8eV. However, this should be called "optical bandgap" (the electronic bandgap is around 2.6eV). Also, in the same caption they mention non-existing "Fig.S8f", which should be Fig.S9f.

2) Page 7 in the main text: ".. and on/off ratio over 10⁵ (Fig. 3a)." From Fig.3a the on/off ratio is not clear (guess should be Fig.2b, inset).

3) Fig.S8 could be discussed in more detail. Also, this figure is mentioned in the main text for the first time only after Fig.S9.

4) Most font sizes in Fig.1 should be increased.

5) Fig.1c,d and Fig.3d,e,f. The colors should be turned around (blue for the CB and red for the VB).

6) L and W of the devices should be mentioned in the text.

Reviewer #1 (Remarks to the Author):

The authors present experimental techniques to 1) produce wrinkle-free hBN layers between metals and MoS₂ and to 2) show that these layers can serve as tunneling barriers that suppress Schottky barrier formation and Fermi level pinning in MoS₂ based transistors. Their tunneling contact devices show gate tunable doping and ambipolar diode behavior.

To my understanding, these findings are indeed novel and of great importance for TMD based device designs. Being a “pure theoretician”, I cannot judge the experimental techniques presented in this work. However, I agree with the authors that their findings of how to fully gate control the effective charge density in MoS₂ based FETs has high impact on the overall TMD-based device literature. In that light, I believe the publication of this work in Nature Communications is appropriate after mandatory changes are done:

We thank Reviewer #1 for his/her positive comment. We present here answers to his/her questions in the following, and also have corrected the manuscript correspondingly. Corrections are highlighted in the SI or main text in the revised version of our manuscript.

I have some concerns about the theoretical section of this work:

It would be interesting to see what effective barrier height the DFT calculations showed for the hBN barrier.

Our answer:

To explore the effective barrier height of 2-layered h-BN, we calculated the average electrostatic potentials along y directions for each model in Fig. R1 (Fig. S9 in the updated SI).

The effective tunneling barrier height ΔV is defined as the potential barrier height above the Fermi level of the metal contact [Nat. Comm., 6, 6181 (2015)], indicated by the blue rectangle in Fig. R1a-b. Two tunnel barriers in series TB1 and TB2 (Fig. R1b) can be seen in the 2-layered h-BN TC-FET model in comparison with the conventional Au-MoS₂ contacted model. As can be seen from Fig.S9b, The maximum tunneling barrier height ΔV_{max} is about 4.05 eV.

Following the referee’s suggestion, we provided relevant discussions in section 4.3 of the Suppl. Info.

Figure R1. Side views of the average electrostatic potentials along y direction for each model. (a) Normal contact, and (b) 2-layered h-BN tunnel contact. The Fermi level is set to zero. The blue rectangular boxes represent the tunnel barriers because of the vacuum gap.

The observed current densities of the tunnel-contact FETs are 2-3 orders of magnitude smaller than the conventional metal contacts. Do (even) thinner barriers show higher current densities with still ambipolar characteristics? Do the theoretical calculations show saturation of the current at higher voltages?

Our answer:

According to the referee's valuable comment, we have calculated the I - V characteristics of 1-layered h-BN TC-FET model, and found that the current density is 2-3 orders of magnitude higher than that of 2-layered h-BN device. However, the rectifying behavior does not prevail under gate voltages of, for example, $V_g=0.0V$, as shown in Fig. R2.

We notice that, experimentally, the chance to have exfoliated mono layer h-BN is extremely low, and the thinnest limit used in this work is 2-layered h-BN. Nevertheless, chemical vapor deposition (CVD) grown mono layer h-BN has been reported previously (Ref. 36. in the main text of our manuscript). Their experimental observation showed no rectifying behavior when CVD grown monolayer h-BN is used as a spacing layer, probably due to a large amount of defects existing in the CVD h-BN.

Figure R2. I-V characteristics in a 1-layered h-BN tunnel contacted FET predicted by *ab initio* simulation under gate voltage $V_g=0.0V$.

On the other hand, Fig. 3i in the main text of our manuscript shows the I - V characteristics of 2-layered h-BN model at $V_g=3$ V. One can see the current densities at bias voltage higher than 0.8 V show a saturation tendency, which is in good agreement with the experimental result. Following the reviewer’s suggestion, we further plot the intensity at higher voltage and the saturation of the current can be still observed up to 1.2 V, which can be found in the Fig. R3. We expect current saturation in even higher bias voltages, although calculation difficulty (poor convergence) at far from equilibrium condition prohibits us from further simulation.

The above discussions have been added in section 4.5 of the Suppl. Info.

Figure R3. I - V characteristics in a 2-layered h-BN tunnel contacted FET. Current saturation can be seen under higher bias voltages predicted by *ab initio* simulation.

To make the DFT results reproducible, however, the authors have to add more information about the relative orientation of the hBN, MoS2 and metal. The relaxation convergence limit alone is not a unique information for the structure setup otherwise. How is the structure setup in the model?

Our answer:

Thanks for the reviewer's valuable suggestion.

In order to achieve the smallest lattice mismatch, we use 2×3 unit cells of Au atoms in (100) orientation, 2×3 unit cells of h-BN and 1×4 unit cells of MoS₂ to set up the device. The relaxed lattice structures of Au, h-BN, MoS₂ and the corresponding lattice constants in x , z directions are shown in Fig.S8a-c in the updated SI (section 4.1). Due to the fact that the band structure of MoS₂ is sensitive to its lattice structure, we parameterize the lattice constants of Au and h-BN solely in x direction to match that of MoS₂. Because the lattice constant in x direction of h-BN is very similar to that of MoS₂, we applied a 0.9% tensile strain of h-BN in the x direction to make it matching with MoS₂. For the Au electrode, we need to apply a 3.4% compression strain in x direction to make it matching with MoS₂, which is acceptable with only slight changes in the band structure of the metal.

The leftmost and rightmost four layers of the Au electrodes in the device are determined via a separate calculation and shifts rigidly relative to each other by the external bias voltage. The rest part is the central scattering region that contains a portion of the Au electrodes, thereby establishing the bonding between the channel and the electrode, the common Fermi level, and the charge neutrality at equilibrium. The infinite open boundary problem is thereby reduced to a proper, self-consistent calculation of the charge density for the finite-sized scattering region.

The above discussions have been added in section 4.1 of the Suppl. Info.

Is that relaxation limit small enough for converged results? How much strain is in the system with the achieved relaxation result. What is the system's size in periodic direction? Is the energy cutoff of 150Ry a converged value?

Our answer:

The geometries are optimized until all residual force tolerance is smaller than 0.05 eV/Å. This relaxation limit follows the previous studies on the metal-MoS₂ contacts [IEEE Int. Electron Dev. Meet. 407-410 (2012); Phys. Rev. X 4, 031005 (2014); ACS Nano 9, 869 (2015); Nano Lett. 16, 2234 (2016)]. The optimized geometry of the three materials is in line with the above results.

We believe the relaxation limit used here is small enough in order to obtain converged results. As mentioned in the answer of the previous question, we change the lattice constants in x direction of Au and h-BN to match the MoS₂ due to the sensitive influence of lattice structure on the band structure of MoS₂.

Because the lattice constant in the x direction of h-BN is very similar to that of

MoS₂, we just apply a 0.9% tensile strain of h-BN in *x* direction to make it match up with MoS₂, while a 3.4% compression strain in the *x* direction is needed for the Au electrode to match with MoS₂. Therefore, the distribution of strains in the system is mainly localized to the Au electrodes after relaxation. Although the compression strain will affect the electronic structure of Au electrode, the general metallicity cannot be changed. Therefore, the strains of Au electrodes have little effect on our calculation about the device's electronic transport properties.

The system's size is 0.543 nm in the periodic *x*-direction (Fig.S8b), while the central scattering region is of 9.8 nm in the *z* direction including MoS₂ channel, BN tunnel contact, and Au electrode extension.

The energy cutoff of 150 Ry was adopted in previous study of the Metal-MoS₂-Metal device [Phys. Rev. B 89, 245403 (2015)]. In order to judge whether the energy cutoff of 150 Ry is a converged value, calculations with larger energy cutoff (180 Ry and 200 Ry) were also tested. The calculated results such as band gap of MoS₂, transmission spectrum of normal- and tunnel contacted device are similar to that of 150 Ry. Therefore, an energy cutoff of 150 Ry is eventually used in this work to achieve a balance between calculation efficiency and accuracy.

Following referee's suggestion, we provided more discussions regarding the relaxation and cutoff energy details in section 4.2 of the Suppl. Info.

I am also missing a justification for the chosen LDA functional. If that was for convenience/numerical load reduction only (irrespective of possibly wrong band gap results) but to get qualitative comparisons, it should be mentioned in the paper.

Our answer:

In this work, we have chosen LDA with HGH to perform our simulations. LDA functional had been proved to yield rather accurate band structures of monolayer MoS₂ and/or other TMDCs [J. Phys. Chem. C 116, 21556 (2012); J. Phys. Chem. C 116, 8983 (2012); Phys. Rev. B 88, 195420 (2013); Phys. Rev. X 4, 031005 (2014); Nano Lett. 14, 1714 (2014); Nano Lett. 15, 4616 (2015); Nanotechnology 27, 105702 (2016)]. Our results show that LDA with HGH gives a direct band gap of 1.8 eV for the monolayer MoS₂, which is consistent with results from experiments [Phys. Rev. Lett. 105, 136805 (2010)].

Indeed, as also pointed out by Reviewer #2, for example, LDA with quasi-particle corrections will give more accurate band structures (band gap of 2.3-2.6 eV in monolayer MoS₂, instead of the 1.8 eV, which is close to its optical band gap, calculated here). However, this correction will consume significantly the computational power but will not affect the general transport behavior, as have been shown in previous studies [Phys. Rev. B 88, 195420 (2013); Phys. Rev. X 4, 031005

(2014); Nano Lett. 14, 1714 (2014)].

Finally, as a result of balancing simulation efficiency and accuracy, we have used LDA with HGH in this work.

Following referee's suggestion, we provided discussions regarding the chosen LDA functional in section 4.4 of the Suppl. Info.

Reviewer #2 (Remarks to the Author):

The authors report a very interesting realization of MoS₂ TC FETs which allows to suppress the Schottky barriers and Fermi level pinning by using tunnel thin hBN underneath S/D contacts. Under some bias conditions this brings to the p-FET performance, which has been previously considered as not typical for MoS₂ FETs. The results are properly verified by comparing the performance of MoS₂ TC FETs and conventional (SB) MoS₂ devices fabricated on the same chip.

We thank Reviewer #2 for his/her positive comment.

He/she has concerns about some important issues that will help us in strengthening our findings. We now answer his/her comments in the following text.

However, the paper can be considerably improved by addressing the following points before publishing in Nature Communications.

Major comments:

1) The use of tunnel-thin hBN layer used for the passivation of the S/D contacts introduces an additional MoS₂/hBN interface into the device architecture. This may lead to additional reliability issues due to charge trapping by hBN defects which can exchange charges with the MoS₂ channel. Thus I would suggest the authors to verify the presence of the hysteresis on the Id-Vg and, most importantly (to understand the contribution of the tunnel layer), the Id-Vds characteristics of MoS₂ TC-FETs. Comparison of the hysteresis results for TC and standard (SB) MoS₂ FETs could be provided in the Supporting Information.

Our answer:

To verify the reliability issues raised by Reviewer #2, a side-by-side comparison of hysteresis in both $I_{ds}-V_g$ and $I_{ds}-V_{ds}$ between the TC-FET and standard (SB) MoS₂ FET has been carried out, as shown in Fig. R4 (see also Fig. S13).

We found that, our TC-FET show as small hysteresis as that of normal MoS₂ FET, in both field effect curves and IV curves, as shown in the figure below:

Figure R4. (a)-(c) Hysteresis measurements of field effect and IV curves for MoS₂ TC-FET, with (c)-(e) the gate leakage current I_{leak} recorded corresponding to each transport curve above. (f)-(h) Hysteresis measurements of field effect and IV curves for normal MoS₂ FET, with (i)-(k) the gate leakage current I_{leak} recorded corresponding to each transport curve above.

The above data show side-by-side the comparison of the hysteresis of field-effect

and I/V curves for TC-FET and normal MoS₂ FET, together with their corresponding gate leakage current (I_{leak}). It has been added into the SI in our revised manuscript. And a paragraph is added in the SI as follows:

“The use of tunnel-thin h-BN layer intercalated between the S/D contacts and MoS₂ channel induces an additional interface in the device. It is therefore important to clarify if extra hysteresis is caused either in the field effect or I/V curves. Here we show side-by-side comparison between TC-FET and normal MoS₂ FET their hysteresis measurements in Fig. S13.

It can be seen that the TC-FET show as small hysteresis as that of normal MoS₂ FET, in both field effect curves and I/V curves. This can rule out, in future applications, the reliability issues which could be caused by charge trapping between the h-BN and MoS₂ interfaces.”

Correspondingly, a brief sentence in the main text is added in Page.11 to refer to the hysteresis measurements shown in the SI, as following:

“Moreover, from the application point of view, the thin h-BN layer intercalated between the metal contacts and MoS₂ channel can cause charge trapping that may lead to reliability problems. We rule out this possibility based on the hysteresis measurements, as shown in Fig. S13.”.

2) In the end of page 5 the authors say that the leakage currents (through the thin hBN layer) are limited by 100pA. However, in Fig.S6 the tunnel current can be as high as 12nA. How one should understand this? Also, it seems that both the useful (field-effect) and leakage currents are called I_{ds} in the graphs and in the text. To avoid any confusion, can the authors call the leakage current as I_{leak} (for instance)?

Our answer:

We are sorry for the confusion. The current plotted in Fig. S6 in our previous manuscript is the field-effect current up to 12 nA, not the gate leakage current.

We have denoted in the revised manuscript the leakage current as I_{leak} , as suggested by Reviewer #2. In particular, I_{leak} (leakage current from the bottom gate) for both field-effect and I/V curves are also shown correspondingly in Fig. S13.

The corrected text (Page 5) in the manuscript is summarized (which has been moved into SI as suggested by Reviewer #2) as below:

“We recorded the gate leakage current (I_{leak}) from the bottom graphite gate concomitant with transport measurement...”.

3) A detailed discussion of the Figures from the Supporting Information (e.g. pages 5-6) makes the paper less comprehensive. I would suggest to move these details into the SI (where these figures are provided), while leaving only brief references to the SI in the main text.

Our answer:

We fully agree with Reviewer's comments.

The discussion has been moved to SI, and only one brief reference is left in main text. As highlighted by the red color in Page 5, also listed below:

“A detailed comparison of transfer curves between MoS₂ normal FET and TC-FET, as well as data from various samples are given in the Suppl. Info.”.

4) One of the main advantages of MoS₂ compared to other TMDs is large I_{ON} values. However, in the TC MoS₂ FETs I_{ON} is nearly 3 orders of magnitude lower than in conventional MoS₂ devices (nA vs. uA). As I understand, this is because of "screening" of the applied V_d due to the presence of the insulator under the S/D contacts (e.g. 1V V_{ds} in TC FETs could be equivalent, for instance, to 0.1V in conventional devices). At the same time, the major problem of these devices seems to be a considerable limitation on V_d operation range (i.e. one can not apply very high V_{din} order to avoid damage of thin hBN layer). Can the authors comments on this and estimate "critical" V_d for the hBN thickness used?

Our answer:

The reviewer points out that the working-state current (I_{ON}) in our devices are 2-3 orders of magnitude lower than that in conventional MoS₂ device. Indeed, this is one of the major limits in our prototype devices. While presenting powerful gate-tunable *pn-to-np* rectifying function, it sacrifices I_{ON} significantly.

However, there might be ways to overcome this problem. For example, one can put multiple devices in parallel, or replace the conducting channel with other TMDCs. This will definitely be one of the major goals in our future studies.

To give an estimated 'critical' V_d for the h-BN thickness used in our study, we would say, in most of the devices that we studied, for a V_g range of -7 to +7 V, V_d is limited within 2.5 V, otherwise there would be gate leakage happening. As already described in the main text, the measurements were carried out with “all V_g and V_{ds} in the measurements were pushed to the limit which keeps gate leakage negligible”.

Minor Comments:

1) In Fig.S9 caption the authors say that the bandgap of monolayer MoS2 is 1.8eV. However, this should be called "optical bandgap" (the electronic bandgap is around 2.3-2.6eV). Also, in the same caption they mention non-existing "Fig.S8f", which should be Fig.S9f.

Our answer:

We thank very much for the Reviewer #2's comments. The term "bandgap" has been corrected in Fig.S9 as "optical bandgap". In the LDA simulations if one takes into account the quasi-particle excitation, the electronic band gap is indeed around 2.3-2.6 eV. Limited by the large computational system, in this work we used conventional LDA functions without quasi-particle corrections. However, this should not affect the general transport behaviors simulated, as have been shown in previous studies [Phys. Rev. B 88, 195420 (2013); Phys. Rev. X 4, 031005 (2014); Nano Lett. 14, 1714 (2014)].

The non-existing "Fig.S8f" is now corrected. In the new SI, it is plotted as "Fig.S10d".

2) Page 7 in the main text: ".. and on/off ratio over 10^5 (Fig. 3a)." From Fig.3a the on/off ratio is not clear (guess should be Fig.2b, inset).

Our answer:

'Fig. 3a' has been corrected into 'Fig. 2b'. As highlighted by red color in the maintext.

3) Fig.S8 could be discussed in more detail. Also, this figure is mentioned in the main text for the first time only after Fig.S9.

Our answer:

The index of figures has been corrected and updated along with new figures in the new SI.

A new paragraph of discussion for Fig.S8 (now indexed as Fig. S12 in the revised manuscript) is also added, as highlighted by red color.

4) Most font sizes in Fig.1 should be increased.

Our answer:

We thank the reviewer for his/her kind suggestions. The modified figure has been updated in the revised manuscript with a bigger font size.

5) *Fig.1c,d and Fig.3d,e,f. The colors should be turned around (blue for the CB and red for the VB).*

Our answer:

We thank the reviewer for his/her suggestions. The colors are now turned around for both Fig. 1c-d and for Fig. 3d-f.

6) *L and W of the devices should be mentioned in the text.*

Our answer:

L and W of the devices are now mentioned in the main text, in Page 5, as indicated by red colored text:

“Typical width of the electrodes are around 1 μm , with the MoS_2 channel beneath having dimensions $L \times W$ of $1 \mu\text{m} \times 1 \sim 5 \mu\text{m}$ for the tested devices.”.

Reviewers' comments:

Reviewer #1 (Remarks to the Author):

The authors have thoroughly clarified my questions and concerns. The updated manuscript can be published as is.

Reviewer #2 (Remarks to the Author):

The authors made a considerable progress in improvement of the manuscript quality. However, there are still some points which require further clarification before publishing:

1) a) The authors conclude that "TC-FET show as small hysteresis as that of normal MoS2 FET". However, the hysteresis width is typically dependent on the sweep rate. In particular, small hysteresis can be observed simply because of very fast sweep rate. Thus, the authors should at least mention the sweep rate used for their measurements in Fig.S13, or, better, compare the dependences of the hysteresis width versus measurement frequency $f=1/(N*\text{step})$ for two types of devices (see e.g. Y. Illarionov et al, 2D Materials, 035004, 2016 and Y. Illarionov et al, ACS Nano, 9543, 2016).

b) The Id-Vg and Id-Vd curves measured using different sweep directions and corresponding arrows could be made in different colour. Also, the current axis should be made in log-scale, in order to make the hysteresis around V_{th} more visible (by the way, now it seems that for normal devices the hysteresis of Id-Vg curves is larger).

2) Description of Fig. S6 is still confusing. In their response the authors mention that I_{ds} is a field-effect current (i.e. through the MoS2 channel). However, in the description it is called tunneling current. Indeed, the tunneling current should depend on the area of tunnel contacts. But the current plotted in Fig.S6 is definitely field-effect(I_{ds}) but not the tunneling current. Furthermore, in Figure S6 caption the authors mention "the I_{ds} -Vg characteristics of left/right tunnel electrodes". The Reviewer thinks that this should be the I_{ds} -Vg characteristics of TC-FETs with left/right tunnel electrodes used as source and drain. Then the difference in I_{ds} is not due to different area of the electrodes, but e.g. due to different dimensions ($L*W$) of left and right TC-FETs.

3) The authors removed unnecessary description of supplementary figures from the main text. However, now most figures from the Supplementary Information (SI) are not mentioned in the text at all. The Reviewer thinks that every figure from the SI should be briefly mentioned in the text (e.g. "for more details see Fig. S1 in the Suppl.Inf.").

Reviewer #1 (Remarks to the Author):

The authors have thoroughly clarified my questions and concerns. The updated manuscript can be published as is.

We thank the reviewer very much.

Reviewer #2 (Remarks to the Author):

The authors made a considerable progress in improvement of the manuscript quality.

We thank the reviewer very much for his/her comments.

However, there are still some points which require further clarification before publishing:

*1) a) The authors conclude that "TC-FET show as small hysteresis as that of normal MoS₂ FET". However, the hysteresis width is typically dependent on the sweep rate. In particular, small hysteresis can be observed simply because of very fast sweep rate. Thus, the authors should at least mention the sweep rate used for their measurements in Fig.S13, or, better, compare the dependences of the hysteresis width versus measurement frequency $f=1/(N*tstep)$ for two types of devices (see e.g. Y. Illarionov et al, 2D Materials, 035004, 2016 and Y. Illarionov et al, ACS Nano, 9543, 2016).*

Our answer:

We fully agree with the reviewer that the hysteresis width is typically dependent on the sweep rate. Here, we used a sweep rate of 20 mV/s (for all sweeping of V_{gate} or V_{ds} up and down) in Fig. S13. The sweeping rate is given in Page 15 in the updated Suppl. Info.

As suggested by the reviewer, we are inspired to carry out a comprehensive study of our TC-FET as shown in references (such as Y. Illarionov et al, 2D Materials, 035004, 2016 and Y. Illarionov et al, ACS Nano, 9543, 2016). The relative investigations inspired by the reviewer will be further investigated in our future publication.

To strengthen on this point, we have added the two references given by Reviewer #2, and added a brief discussion in page 11 in the revised main text, as below:

“Moreover, stability and reliability in 2D materials based devices have been a timely topic [40, 41], which is crucial from the application point of view. For example, the thin h-BN layer intercalated between the metal contacts and MoS₂ channel can cause extra charge trapping that may lead to inferior reliability as compared to

conventional metal contacted MoS₂ FETs. We rule out this possibility based on the hysteresis measurements, as shown in Fig. S13.”

b) The I_d - V_g and I_d - V_d curves measured using different sweep directions and corresponding arrows could be made in different colour. Also, the current axis should be made in log-scale, in order to make the hysteresis around V_{th} more visible (by the way, now it seems that for normal devices the hysteresis of I_d - V_g curves is larger).

Our answer:

Following the reviewer’s suggestion, we have re-plotted the curves with a log-scale current axis in different colors for different sweeping directions, as shown in Fig.S13 in the revised manuscript.

2) Description of Fig. S6 is still confusing. In their response the authors mention that I_{ds} is a field-effect current (i.e. through the MoS₂ channel). However, in the description it is called tunneling current. Indeed, the tunneling current should depend on the area of tunnel contacts. But the current plotted in Fig.S6 is definitely field-effect(I_{ds}) but not the tunneling current. Furthermore, in Figure S6 caption the

authors mention "the I_{ds} - V_g characteristics of left/right tunnel electrodes". The Reviewer thinks that this should be the I_{ds} - V_g characteristics of TC-FETs with left/right tunnel electrodes used as source and drain. Then the difference in I_{ds} is not due to different area of the electrodes, but e.g. due to different dimensions ($L \times W$) of left and right TC-FETs.

Our answer:

We fully agree with the reviewer and we thank him/her for pointing that out. Fig.S6 shows a field effect (I_{ds} - V_g characteristics) of TC-FETs with left/right tunnel electrodes used as source and drain. We have corrected the relative descriptions in the Suppl. Info (highlighted in blue color in Page 7 in the Suppl. Info.), as below:

"We find that the amplitude of I_{ds} - V_g characteristics is dependent on the area of tunnel contacts, as well as the dimensions ($L \times W$) of the channel of the TC-FET. As shown in Fig. S6, the aspect ratio of left and right TC-FETs are 4~5 times difference, and the areas of the TC contacts of them are also 4~5 times difference, which explains the resulted magnitude of ~25 times change in the I_{ds} - V_g characteristics."

We thank again the referee for bring us to a more precise understanding in this issue.

3) The authors removed unnecessary description of supplementary figures from the main text. However, now most figures from the Supplementary Information (SI) are not mentioned in the text at all. The Reviewer thinks that every figure from the SI should be briefly mentioned in the text (e.g. "for more details see Fig. S1 in the Suppl. Info.").

Our answer:

We have made corrections in the main text in the revised manuscript. All figures from the SI are now mentioned in the text (highlighted as blue color in the new version in the main text, in Page 3, Page 5, Page 8, and page 11).

REVIEWERS' COMMENTS:

Reviewer #2 (Remarks to the Author):

There are no further comments from my side. The manuscript is suitable for publication.

Reviewer 2: Dr. Yu.Yu. Illarionov